# OpenReview forum: "Deep Positive-Unlabeled Anomaly Detection for Contaminated Unlabeled Data"
_ICLR.cc/2026/Conference — Submitted to ICLR 2026_

### Official Review · Reviewer_4853 · 2025-10-21

**Soundness:** 2
**Presentation:** 3
**Contribution:** 2
**Rating:** 2
**Confidence:** 4

**Summary:**

This paper proposes a novel method for semi-supervised anomaly detection where training data consisits unlabeled data and labeled anomaly data. Specially, the method assumes the distribution of unlabeled dataset is a combination of normal and anomalous data distribution, and then uses PU learning to minimize the anomaly score of normal data in the unlabeled dataset while maximizing the anomaly score of anomalous data in the unlabled dataset.

**Strengths:**

- The paper is well written and easy to follow.
- The proposed method is simple yet effective.

**Weaknesses:**

- Lcak of novelty. The method seems like a direct use of the objective function $\hat{R}_{abs-PU}(g)$ in [1].
- No theoritical analysis is provided.
- The baselines are not novel enough. Many AD methdos for the comtinated datasets and semi-supervised AD are proposed recently. For instance, [2], [3] and [4].
- The evaluation is merely on image datasets. More evaluation on popular benchmark like ADbench [5] should be included. Also, it seems that the anomaly ratio in unlabeled data is fixed to 10%, evaluation on datasets with different anomaly ratio  should be included to evaluate the effectiveness of the proposed method.
- In experiment part, a validation set is created, but usually in AD, there is no such validation set. What is the reason for such a validation set, and is it reasonable?
- In hyperparameter analysis, it seems that the performance of method is sensitive to change of $\alpha$ in some datasets. In experimental setup,  $\alpha$ is set to be the anomaly ratio. But in real application, the anomaly ratio usually can not be obtained which restricts the use of this method.
- The experimental setup is weak. In experimental setup, you mentioned 'select one normal class from the remaining 9 classes', but in previous research like [6], each class is treated as normal once a time and 10 experiments is conducted for every dataset. Also, what is the reason for subsampling 2000 samples from the original test set as your test set?

[1] Hammoudeh Z, Lowd D. Learning from positive and unlabeled data with arbitrary positive shift[J]. Advances in Neural Information Processing Systems, 2020, 33: 13088-13099.

[2] Durani, Walid, et al. "Weakly Supervised Anomaly Detection via Dual-Tailed Kernel." Forty-second International Conference on Machine Learning.

[3] Pang, Guansong, et al. "Deep weakly-supervised anomaly detection." Proceedings of the 29th ACM SIGKDD Conference on Knowledge Discovery and Data Mining. 2023.

[4] Xu, Hongzuo, et al. "RoSAS: Deep semi-supervised anomaly detection with contamination-resilient continuous supervision." Information Processing & Management 60.5 (2023): 103459.

[5] Han, Songqiao, et al. "Adbench: Anomaly detection benchmark." Advances in neural information processing systems 35 (2022): 32142-32159.

[6] Ruff, Lukas, et al. "Deep semi-supervised anomaly detection." arXiv preprint arXiv:1906.02694 (2019).

**Questions:**

See weakness.

---

> ### Author Response · Authors · 2025-11-21
> **Author response to Reviewer 4853 (1/2)**
>
> Thank you for your insightful comments, which we shall address below.
>
>
> > Lcak of novelty. The method seems like a direct use of the objective function in [1].
>
> As you commented, our framework integrates unbiased PU learning with deep detectors, but this integration is non-trivial and hence has not been explored before. Conventional PU learning is originally designed for binary classification, and when it is directly applied to anomaly detection, it cannot detect unseen anomalies, as shown in Figure 1(a).
>
> Therefore, it is necessary to introduce formulations such as Eqs. (4) and (15), which enable an anomaly detector to be interpreted as a binary classifier while preserving the ability to detect both seen and unseen anomalies. In particular, the formulation in Eq. (4) allows deep detectors to be interpreted as probabilistic binary classifiers, which enables us to directly inherit the desirable properties of unbiased PU learning.
>
> We believe that, through this contribution, our framework that integrates deep detectors with unbiased PU learning is both novel and practically important for the semi-supervised anomaly detection problem.
>
>
> > No theoritical analysis is provided.
>
> Our approach inherits the theoretical properties of unbiased PU learning. In particular, PU learning methods based on the absolute-value correction, such as Eq. (13), have been shown to yield a consistent estimator for arbitrary functions (Hammoudeh & Lowd, 2020). Consequently, even in our setting where the anomaly detector is interpreted as a probabilistic binary classifier, as in Eq. (3), this property still holds, and we can show that the ideal anomaly detector is learned as the number of data points increases. We will include this analysis in the revised paper.
>
>
> > The baselines are not novel enough. Many AD methdos for the comtinated datasets and semi-supervised AD are proposed recently. For instance, [2], [3] and [4].
>
> Thank you for the suggestion. In our problem setting, we regard SOEL as the state-of-the-art method, which was presented around the same time as [2,3,4]. For example, the method in [3] assumes that the unlabeled data are almost entirely normal, and therefore does not match our setting where the unlabeled data are contaminated. We have compared the proposed method with various approaches, including SOEL as the state-of-the-art, and confirmed that it achieves the best performance. We will include the suggested paper in the related work section and add a discussion of its relationship to our proposed method in the revised paper.
>
>
> > The evaluation is merely on image datasets. More evaluation on popular benchmark like ADbench [5] should be included. Also, it seems that the anomaly ratio in unlabeled data is fixed to 10%, evaluation on datasets with different anomaly ratio should be included to evaluate the effectiveness of the proposed method.
>
> Since the AEs and DeepSVDD used in this paper are known to achieve strong performance especially for image anomaly detection, we evaluate our approach on a wide range of image datasets: from simple image datasets such as MNIST, FashionMNIST, SVHN, CIFAR-10, and CIFAR-100, to real-world medical image datasets such as Path, OCT, and Tissue, and furthermore on MVTecAD, which is a current standard image benchmark for anomaly detection as of 2025, in Appendix E. On all of these datasets, the proposed method achieves the best performance.
>
> We are also considering extending the proposed method to tabular data. Our framework can be applied as long as the loss function of the base anomaly detector is non-negative and differentiable. In addition to the AE and DeepSVDD used in this paper, it should also be applicable to more recent anomaly detectors for tabular data such as [a,b]. We leave the application of our framework to these methods for future work.
>
> In Figure 4 of Appendix C, we compare the performance of PUSVDD and SOEL under different true contamination rates. Even when the true contamination rate changes, PUSVDD consistently outperforms SOEL.
>
> [a] Qiu, Chen, et al. "Neural transformation learning for deep anomaly detection beyond images." International Conference on Machine Learning. PMLR, 2021.
>
> [b] Shenkar, Tom, and Lior Wolf. "Anomaly detection for tabular data with internal contrastive learning." International Conference on Learning Representations. 2022.
>
>
> > In experiment part, a validation set is created, but usually in AD, there is no such validation set. What is the reason for such a validation set, and is it reasonable?
>
> When training deep models with stochastic gradient methods, it is common practice to create a validation set and perform early stopping to prevent over-fitting. This practice is not specific to anomaly detection, is easy to implement since it only requires splitting the training data, and is an effective way to mitigate over-fitting. Note that there is no need to use fully labeled data (normal and anomaly) for the validation data.

---

> > ### Author Response · Authors · 2025-11-21
> > **Author response to Reviewer 4853 (2/2)**
> >
> > > In hyperparameter analysis, it seems that the performance of method is sensitive to change of $\alpha$ in some datasets. In experimental setup, $\alpha$ is set to be the anomaly ratio. But in real application, the anomaly ratio usually can not be obtained which restricts the use of this method.
> >
> > Thank you for the helpful comment. By combining an estimation method for the contamination rate $\alpha$ with our proposed method, we can perform end-to-end anomaly detection. Specifically, we propose to combine PUSVDD with class prior estimation (CPE), an $\alpha$-estimation method presented by Christoffel et al. (2016). Since CPE provides a closed-form estimate of $\alpha$, it is very simple and computationally efficient. The procedure is as follows:
> >
> > 1. First, as described in Section 5.3, we pre-train the base DeepSVDD.
> > 2. Next, for the unlabeled data and anomaly data, we extract features using DeepSVDD and apply CPE to estimate $\alpha$.
> > 3. Finally, we train PUSVDD using the estimated $\alpha$.
> >
> > For MNIST and FashionMNIST, when we estimated $\alpha$ using steps 1–2 above, the resulting estimates were $\hat{\alpha}=0.048$ for MNIST and $\hat{\alpha}=0.056$ for FashionMNIST. In Figure 2, we compare the performance of PUSVDD and SOEL while varying $\alpha$. As shown in the figure, for most values of $\alpha$, PUSVDD outperforms SOEL that uses the true $\alpha$. Notably, at $\hat{\alpha}=0.048$ on MNIST and $\hat{\alpha}=0.056$ on FashionMNIST, PUSVDD still achieves higher performance than SOEL that uses the true $\alpha$. Therefore, even if there is some estimation error in $\alpha$ when using existing $\alpha$-estimation methods (Menon et al., 2015; Ramaswamy et al., 2016; Jain et al., 2016; Christoffel et al., 2016), PUSVDD can still achieve higher performance than SOEL.
> >
> > In addition, in Figure 4 of Appendix C, we compare the performance of PUSVDD and SOEL under different true contamination rates. Even when the true contamination rate changes, PUSVDD consistently outperforms SOEL. These results indicate that even when $\alpha$ deviates from the true contamination rate, the proposed method can still achieve higher performance than existing methods.
> >
> >
> > > The experimental setup is weak. In experimental setup, you mentioned 'select one normal class from the remaining 9 classes', but in previous research like [6], each class is treated as normal once a time and 10 experiments is conducted for every dataset. Also, what is the reason for subsampling 2000 samples from the original test set as your test set?
> >
> > We apologize for the confusion. What we did was to select each class as the normal class in turn, run the experiment five times for that class, and then repeat this for all classes. We then report the average results in Tables 1 and 2. The sentence at the beginning of Section 5.4, "We showed the average of the AUROC scores for all normal classes," is intended to convey this. We report only the averages in order to save space and to include results on a larger number of datasets.
> >
> > The reason for sampling 2,000 test samples is as follows. Our intention was to make the ratio of normal data : seen anomalies : unseen anomalies in the test set equal to 2:1:1. Under this constraint, the total number of test samples is determined by the maximum number of anomalies available in the dataset. For the datasets used in this paper, setting the test set size to around 2,000 allowed us to satisfy this condition across all of them, which is why we adopted this design.
> >
> > We believe that this experimental setting is very fair and sufficient to demonstrate the performance of the proposed method.

---

> ### Comment · Reviewer_4853 · 2025-11-28
>
> I thank the authors for their detailed responses, which have resolved most of my concerns. I will raise my score to **4**.
>
> With the expectation of seeing the following additional results, I would like to further raise my score:
>
> - Could the authors provide complete results on each dataset when treating **each individual class** as normal data ?
> - Could the authors compare the proposed approach with **more recent baselines**? Since the most recent baseline considered is **SOEL (ICML 2023)**, several newer methods have been published at recent **ICML, ICLR, and NeurIPS**. Including comparisons on datasets such as **CIFAR-10**  and/or **SVHN**, if applicable, would strengthen the empirical support.
>
> I look forward to these additional evaluations and believe they would further substantiate the contribution of the work.

---

> > ### Author Response · Authors · 2025-11-28
> >
> > Thank you very much for your response and for the helpful suggestions. We will do our best to carry out the additional experiments within the remaining time and will report back with the results as soon as they become available.

---

> ### Author Response · Authors · 2025-12-02
> **Author response to additional comments from Reviewer 4853 (1/2)**
>
> > Could the authors provide complete results on each dataset when treating each individual class as normal data ?
>
> As you suggested, for PUSVDD and SOEL, we provide results on each dataset when treating each individual class as normal data. The experimental results are shown below, where the “Class” column indicates the normal class and AUC is used as the evaluation metric. We ran each experiment five times with different random seeds, and used boldface to indicate the best results and statistically non-different results according to a pairwise t-test with a significance level of 5%.
>
> ---
>
> MNIST:
>
> | Class | SOEL              | PUSVDD            |
> | ----- | ----------------- | ----------------- |
> | 1     | 0.995 ± 0.001     | **0.999 ± 0.000** |
> | 2     | **0.927 ± 0.047** | **0.988 ± 0.003** |
> | 3     | **0.986 ± 0.009** | **0.996 ± 0.001** |
> | 4     | **0.982 ± 0.013** | **0.997 ± 0.001** |
> | 5     | 0.935 ± 0.024     | **0.989 ± 0.009** |
> | 6     | **0.964 ± 0.005** | **0.967 ± 0.013** |
> | 7     | 0.974 ± 0.006     | **0.996 ± 0.002** |
> | 8     | 0.939 ± 0.023     | **0.987 ± 0.004** |
> | 9     | 0.960 ± 0.010     | **0.987 ± 0.004** |
>
> ---
>
> FashionMNIST:
>
> | Class | SOEL              | PUSVDD            |
> | ----- | ----------------- | ----------------- |
> | 1     | 0.983 ± 0.003     | **0.993 ± 0.002** |
> | 2     | **0.930 ± 0.019** | **0.952 ± 0.010** |
> | 3     | **0.911 ± 0.007** | **0.930 ± 0.015** |
> | 4     | 0.920 ± 0.008     | **0.945 ± 0.014** |
> | 5     | 0.985 ± 0.004     | **0.993 ± 0.002** |
> | 6     | **0.750 ± 0.026** | **0.763 ± 0.019** |
> | 7     | **0.994 ± 0.001** | **0.996 ± 0.001** |
> | 8     | 0.960 ± 0.002     | **0.992 ± 0.001** |
> | 9     | 0.989 ± 0.002     | **0.996 ± 0.001** |
>
> ---
>
> SVHN:
>
> | Class | SOEL              | PUSVDD            |
> | ----- | ----------------- | ----------------- |
> | 1     | **0.783 ± 0.040** | **0.850 ± 0.048** |
> | 2     | 0.760 ± 0.014     | **0.801 ± 0.027** |
> | 3     | **0.698 ± 0.032** | **0.682 ± 0.029** |
> | 4     | **0.769 ± 0.033** | **0.788 ± 0.025** |
> | 5     | **0.765 ± 0.012** | **0.799 ± 0.022** |
> | 6     | **0.705 ± 0.013** | **0.690 ± 0.018** |
> | 7     | **0.755 ± 0.042** | **0.771 ± 0.026** |
> | 8     | **0.671 ± 0.022** | **0.658 ± 0.034** |
> | 9     | **0.694 ± 0.031** | **0.675 ± 0.014** |
>
> ---
>
> CIFAR10:
>
> | Class | SOEL              | PUSVDD            |
> | ----- | ----------------- | ----------------- |
> | 1     | **0.793 ± 0.066** | **0.824 ± 0.030** |
> | 2     | 0.712 ± 0.012     | **0.724 ± 0.010** |
> | 3     | **0.724 ± 0.054** | **0.795 ± 0.026** |
> | 4     | **0.731 ± 0.096** | **0.792 ± 0.015** |
> | 5     | 0.790 ± 0.029     | **0.843 ± 0.025** |
> | 6     | **0.856 ± 0.016** | **0.852 ± 0.023** |
> | 7     | 0.780 ± 0.032     | **0.852 ± 0.016** |
> | 8     | **0.751 ± 0.015** | **0.766 ± 0.018** |
> | 9     | **0.782 ± 0.031** | **0.805 ± 0.023** |
>
> ---
>
> Path:
>
> | Class | SOEL              | PUSVDD            |
> | ----- | ----------------- | ----------------- |
> | 1     | **0.965 ± 0.051** | **0.996 ± 0.005** |
> | 2     | **0.526 ± 0.126** | **0.554 ± 0.123** |
> | 3     | **0.950 ± 0.035** | **0.957 ± 0.007** |
> | 4     | **0.737 ± 0.031** | **0.733 ± 0.068** |
> | 5     | **0.741 ± 0.035** | **0.751 ± 0.096** |
> | 6     | **0.812 ± 0.082** | **0.859 ± 0.086** |
> | 7     | 0.803 ± 0.030     | **0.874 ± 0.023** |
> | 8     | 0.797 ± 0.086     | **0.922 ± 0.029** |
>
> ---
>
> Tissue:
>
> | Class | SOEL              | PUSVDD            |
> | ----- | ----------------- | ----------------- |
> | 1     | **0.623 ± 0.017** | **0.622 ± 0.010** |
> | 2     | **0.788 ± 0.052** | **0.843 ± 0.029** |
> | 3     | 0.739 ± 0.029     | **0.806 ± 0.030** |
> | 4     | 0.693 ± 0.016     | **0.749 ± 0.018** |
> | 5     | **0.686 ± 0.035** | **0.722 ± 0.040** |
> | 6     | **0.739 ± 0.026** | **0.716 ± 0.011** |
> | 7     | **0.651 ± 0.037** | **0.655 ± 0.037** |
>
> ---
>
> While SOEL suffers substantial performance degradation for several normal classes, PUSVDD consistently achieves detection performance that is comparable to or better than SOEL across all normal classes. We believe that these results demonstrate the strong practical usefulness of the proposed method.
>
> For completeness, we provide details on the experimental setup. As described in Section 5.1, for MNIST, FashionMNIST, SVHN, CIFAR10, Path, and Tissue, we select each class as the normal class in turn. (For CIFAR100, since the number of classes is too large, we treat the nature-related classes as normal, the human-related classes as seen anomalies, and the people class as the unseen anomaly. In addition, OCT has a pre-defined normal class.) For these six datasets, we index the classes starting from 0. We treat class 0 as the unseen anomaly, then select one of the remaining classes as the normal class, and treat all the remaining classes as seen anomalies. Note that Tables 1 and 2 in the paper correspond to the results obtained by averaging the above per-class results.

---

> ### Author Response · Authors · 2025-12-02
> **Author response to additional comments from Reviewer 4853 (2/2)**
>
> > Could the authors compare the proposed approach with more recent baselines? Since the most recent baseline considered is SOEL (ICML 2023), several newer methods have been published at recent ICML, ICLR, and NeurIPS. Including comparisons on datasets such as CIFAR-10 and/or SVHN, if applicable, would strengthen the empirical support.
>
> Following your suggestion, we compared our approach with Weakly Supervised Anomaly Detection via Dual-Tailed Kernel (WSAD-DTK) [2], which was presented at ICML 2025. As the dataset, we used MVTec AD, which is a standard image benchmark for anomaly detection as of 2025. The experimental setting is the same as in Appendix E. Note that for WSAD-DTK we used a ResNet34 encoder with the same configuration as in PUSVDD and SOEL.
>
> The experimental results are shown below: “all” denotes the detection performance over normal data, seen anomalies, and unseen anomalies altogether; “seen” denotes the detection performance on normal data and seen anomalies; “unseen” denotes the detection performance on normal data and unseen  anomalies. We ran each experiment five times with different random seeds, and used boldface to indicate the best results and statistically non-different results according to a pairwise t-test with a significance level of 5%.
>
> ---
>
> |        | PUSVDD            | SOEL          | WSAD-DTK      |
> | ------ | ----------------- | ------------- | ------------- |
> | all    | **0.793 ± 0.016** | 0.669 ± 0.021 | 0.641 ± 0.029 |
> | seen   | **0.799 ± 0.013** | 0.688 ± 0.021 | 0.642 ± 0.030 |
> | unseen | **0.787 ± 0.020** | 0.649 ± 0.027 | 0.640 ± 0.028 |
>
> ---
>
> Since WSAD-DTK assumes that almost all unlabeled data are normal, it is unable to handle contaminated unlabeled data in the highly contaminated setting considered in our experiments. In contrast, our PUSVDD achieves high anomaly detection performance even under such a highly contaminated setting.
>
> Taken together with the results reported in the main paper and the additional results provided above, these findings show that the proposed method attains strong anomaly detection performance on a variety of datasets and outperforms a recent state-of-the-art semi-supervised method, WSAD-DTK. We therefore believe that the proposed method is a strong approach for semi-supervised anomaly detection with contaminated unlabeled data.

---

### Official Review · Reviewer_qX5f · 2025-10-28

**Soundness:** 2
**Presentation:** 3
**Contribution:** 2
**Rating:** 4
**Confidence:** 3

**Summary:**

This paper proposes a deep positive-unlabeled (PU) anomaly detection framework to address contaminated unlabeled data in semi-supervised anomaly detection. The approach integrates unbiased PU learning (Kiryo et al., 2017) with deep anomaly detectors, specifically autoencoders (PUAE) and DeepSVDD (PUSVDD). The key idea is to approximate anomaly scores for normal data using the relationship: $(1-\alpha)p_N(x) = p_U(x) - \alpha p_A(x)$, where $\alpha$ is the contamination rate. Experiments on 8 datasets demonstrate improvements over existing methods, including SOEL (Li et al., 2023).

**Strengths:**

*Clear presentation:* The paper is well-written with clear mathematical derivations. Figure 1 effectively illustrates the conceptual differences between methods, and the problem formulation is easy to follow.

*Practical problem:* The work addresses the realistic scenario where unlabeled data contain unidentified anomalies, which is common when labeling all anomalies is prohibitively expensive.

*Thorough baseline comparisons:* The paper compares against a comprehensive set of methods spanning unsupervised (IF, AE, DeepSVDD, LOE), semi-supervised (ABC, DeepSAD, SOEL), and PU learning approaches.

*Robustness analysis:* Figure 2 demonstrates that PUSVDD exhibits greater robustness to $\alpha$ misspecification compared to SOEL across multiple datasets, which is a valuable property.

**Weaknesses:**

*1. Limited Technical Novelty*

The core methodology is a direct application of the unbiased PU learning framework from Kiryo et al. (2017). The mathematical framework (Eq. 6-13) follows the standard PU learning derivation without significant modification. The main adaptation is substituting different base loss functions ($\ell_{BCE}$ for PUAE, $\tilde{\ell}_{SAD}$ for PUSVDD) into the established PU learning objective. While applying PU learning to anomaly detection is useful, the technical contribution is primarily an application rather than a methodological innovation.

*2. Missing α Estimation Mechanism*

This represents a fundamental gap that severely limits practical applicability:

- **SOEL (Li et al., 2023) provides an adaptive α estimation mechanism** using importance-weighted estimators that automatically estimate contamination rates from data with theoretical guarantees. This is a core contribution of SOEL.

- **PUSVDD lacks any α estimation implementation.** The paper states that "$\alpha$ can be estimated... in conventional PU learning approaches" (Line 197) but provides neither implementation nor experimental validation.

- **All experiments assume known α = 0.1.** Tables 1-2 compare PUSVDD and SOEL under identical settings where $\alpha$ is given as ground truth. This comparison does not reflect SOEL's capability to automatically estimate $\alpha$, creating an incomplete picture of relative performance.

- **Practical deployment requires knowing α.** Without automatic estimation, users must manually tune $\alpha$ or implement separate estimation methods, significantly limiting the method's practical utility compared to SOEL.

*3. Incomplete Experimental Validation*

**Limited scope of sensitivity analysis:**
- Figure 2 only examines scenarios where true contamination is 10%, varying the algorithm's $\alpha$ parameter from 0.1 to 0.5.
- Missing: experiments with different true contamination rates (e.g., 1%, 5%, 10%, 20%) to validate generalizability.
- A more informative analysis would test multiple combinations of (true α, estimated α) across key datasets.

**No end-to-end evaluation:**
- The paper should demonstrate: α estimation → model training → evaluation, comparing PUSVDD and SOEL when both methods must estimate $\alpha$ from data.
- Current results only show PUSVDD is more robust to α misspecification, but do not demonstrate whether this advantage persists when $\alpha$ is unknown and must be estimated in practice.

*4. Modest Performance Improvements*

While PUSVDD achieves best results in most cases, improvements over SOEL are often marginal:
- CIFAR100: 0.637 vs. 0.633 (0.4% improvement)
- OCT: 0.857 vs. 0.856 (0.1% improvement)
- FashionMNIST: 0.948 vs. 0.936 (1.2% improvement)

Given the lack of α estimation and limited technical novelty, these incremental gains are insufficient to demonstrate substantial advancement over the current state-of-the-art.

**Questions:**

1. **α estimation implementation:** Can you implement at least one α estimation method (e.g., Elkan & Noto 2008, Christoffel et al. 2016, or SOEL's importance-weighted estimator) and provide experimental results? This is essential for demonstrating practical applicability and enabling fair comparison with SOEL.

2. **Extended sensitivity analysis:** Can you provide experiments with varying true contamination rates beyond 10%? For example, prepare datasets with true α ∈ {0.01, 0.05, 0.1, 0.2} and evaluate performance under different estimated α values to comprehensively assess robustness.

3. **End-to-end comparison:** What is the performance when both SOEL and PUSVDD must estimate α from data (i.e., neither method is given the ground truth)? This would provide a fair comparison that accounts for SOEL's adaptive estimation capability and PUSVDD's claimed robustness advantage.

---

> ### Author Response · Authors · 2025-11-21
>
> Thank you for your insightful comments, which we shall address below.
>
>
> > Weakness 1: Limited Technical Novelty
>
> As you commented, our framework integrates unbiased PU learning with deep detectors, but this integration is non-trivial and hence has not been explored before. Conventional PU learning is originally designed for binary classification, and when it is directly applied to anomaly detection, it cannot detect unseen anomalies, as shown in Figure 1(a).
>
> Therefore, it is necessary to introduce formulations such as Eqs. (4) and (15), which enable an anomaly detector to be interpreted as a binary classifier while preserving the ability to detect both seen and unseen anomalies. In particular, the formulation in Eq. (4) allows deep detectors to be interpreted as probabilistic binary classifiers, which enables us to directly inherit the desirable properties of unbiased PU learning.
>
> We believe that, through this contribution, our framework that integrates deep detectors with unbiased PU learning is both novel and practically important for the semi-supervised anomaly detection problem.
>
>
> > Weakness 2: Missing α Estimation Mechanism
>
> Thank you for the helpful comment. As you pointed out, by combining an estimation method for the contamination rate $\alpha$ with our proposed method, we can perform end-to-end anomaly detection. Specifically, we propose to combine PUSVDD with class prior estimation (CPE), an $\alpha$-estimation method presented by Christoffel et al. (2016). Since CPE provides a closed-form estimate of $\alpha$, it is very simple and computationally efficient. The procedure is as follows:
>
> 1. First, as described in Section 5.3, we pre-train the base DeepSVDD.
> 2. Next, for the unlabeled data and anomaly data, we extract features using DeepSVDD and apply CPE to estimate $\alpha$.
> 3. Finally, we train PUSVDD using the estimated $\alpha$.
>
> For MNIST and FashionMNIST, when we estimated $\alpha$ using steps 1–2 above, the resulting estimates were $\hat{\alpha}=0.048$ for MNIST and $\hat{\alpha}=0.056$ for FashionMNIST. In Figure 2, we compare the performance of PUSVDD and SOEL while varying $\alpha$. As shown in the figure, for most values of $\alpha$, PUSVDD outperforms SOEL that uses the true $\alpha$. Notably, at $\hat{\alpha}=0.048$ on MNIST and $\hat{\alpha}=0.056$ on FashionMNIST, PUSVDD still achieves higher performance than SOEL that uses the true $\alpha$. Therefore, even if there is some estimation error in $\alpha$ when using existing $\alpha$-estimation methods (Menon et al., 2015; Ramaswamy et al., 2016; Jain et al., 2016; Christoffel et al., 2016), PUSVDD can still achieve higher performance than SOEL.
>
>
> > Weakness 3: Incomplete Experimental Validation
>
> In Figure 4 of Appendix C, we compare the performance of PUSVDD and SOEL under different true contamination rates. Even when the true contamination rate changes, PUSVDD consistently outperforms SOEL. Moreover, as mentioned above, the results in Figure 2 indicate that even if there is some estimation error in $\alpha$, PUSVDD still achieve higher performance than SOEL that uses the true $\alpha$.
>
>
> > Weakness 4: Modest Performance Improvements
>
> As mentioned above, even if there is some estimation error in $\alpha$, PUSVDD still achieves better performance than SOEL that uses the true $\alpha$. In addition, Table 2 shows approximately a 3% performance improvement on CIFAR-10, Path, and Tissue datasets, while the experiments on MVTecAD in Table 7 of Appendix E achieve more than a 10% improvement. Based on these results, we believe that the proposed method substantially advances the current state of the art.
>
>
> > Questions 1: $\alpha$ estimation implementation & Question 3: End-to-end comparison
>
> Thank you for the suggestion. As discussed in our response to Weakness 2, even when we use the value of $\alpha$ estimated by class prior estimation (Christoffel et al., 2016), the proposed method still achieves higher performance than SOEL that uses the true $\alpha$. Moreover, Figure 2 shows that even if there is some estimation error in $\alpha$, PUSVDD still achieves higher performance than SOEL that uses the true $\alpha$. We believe that these results clearly demonstrate the strong practical utility of the proposed method.
>
>
> > Question 2: Extended sensitivity analysis
>
> Thank you for the suggestion. As mentioned above, in Figure 4 of Appendix C, we compare the performance of PUSVDD and SOEL under different true contamination rates. Even when the true contamination rate changes, PUSVDD consistently outperforms SOEL.

---

> > ### Comment · Reviewer_qX5f · 2025-11-28
> >
> > I appreciate the authors' rebuttal, which has addressed most of my concerns. At this stage, I intend to maintain my original score and would be interested in seeing the feedback from the other reviewers.

---

> > > ### Author Response · Authors · 2025-11-28
> > >
> > > Thank you very much for your thoughtful assessment. Please feel free to let us know if any additional questions or concerns arise during the discussion period.

---

### Official Review · Reviewer_KjdH · 2025-11-02

**Soundness:** 2
**Presentation:** 3
**Contribution:** 2
**Rating:** 4
**Confidence:** 3

**Summary:**

This paper introduces a framework for "deep positive-unlabeled anomaly detection," which focus on the practical and significant challenge of contaminated unlabeled data in semi-supervised anomaly detection. The core idea is to integrate the principles of Positive-Unlabeled (PU) learning with established deep anomaly detection models like Autoencoders (AE) and Deep SVDD. The authors present a solution demonstrated to be empirically effective in handling scenarios where the unlabeled set contains hidden anomalies.

**Strengths:**

The paper tackles a critical and often overlooked limitation of traditional semi-supervised anomaly detection methods, which typically assume that the unlabeled dataset is 'clean' (i.e., contains only normal samples). This assumption rarely holds in real-world settings. By directly addressing the issue of contaminated unlabeled data, this work has strong practical relevance and significant potential for real-world applications.

**Weaknesses:**

While the paper addresses an important problem and the empirical results are promising, I have a few concerns regarding its novelty and theoretical depth that I believe should be addressed to strengthen the contribution.

1. The core contribution appears to be the application of a well-established theoretical framework—unbiased PU learning (Du Plessis et al., 2014; Kiryo et al., 2017)—to existing deep anomaly detection models (AE and DeepSVDD). The key derivations presented in Equations (6)-(11) seem to follow the standard risk reformulation process from the PU learning literature. While the application to this specific problem is new, the work lacks a fundamental innovation to either the PU learning theory itself or the architecture of the deep detectors. As such, the contribution feels more like an effective integration of existing components rather than the proposal of a fundamentally new methodology, which may limit its perceived novelty.
2. The use of an absolute value in the final objective function (Eq. 13) to prevent divergence, while citing prior work, feels somewhat like an ad-hoc fix. This practical necessity might suggest an underlying instability when combining the unbiased PU risk estimator with highly complex models like deep neural networks. The paper would be more impactful if it included a deeper analysis of why this instability occurs and the theoretical implications of this absolute value "patch."

**Questions:**

Please see the weaknesses.

---

> ### Author Response · Authors · 2025-11-21
>
> Thank you for your insightful comments, which we shall address below.
>
>
> > The core contribution appears to be the application of a well-established theoretical framework—unbiased PU learning (Du Plessis et al., 2014; Kiryo et al., 2017)—to existing deep anomaly detection models (AE and DeepSVDD). The key derivations presented in Equations (6)-(11) seem to follow the standard risk reformulation process from the PU learning literature. While the application to this specific problem is new, the work lacks a fundamental innovation to either the PU learning theory itself or the architecture of the deep detectors. As such, the contribution feels more like an effective integration of existing components rather than the proposal of a fundamentally new methodology, which may limit its perceived novelty.
>
> As you commented, our framework integrates unbiased PU learning with deep detectors, but this integration is non-trivial and hence has not been explored before. Conventional PU learning is originally designed for binary classification, and when it is directly applied to anomaly detection, it cannot detect unseen anomalies, as shown in Figure 1(a).
>
> Therefore, it is necessary to introduce formulations such as Eqs. (4) and (15), which enable an anomaly detector to be interpreted as a binary classifier while preserving the ability to detect both seen and unseen anomalies. In particular, the formulation in Eq. (4) allows deep detectors to be interpreted as probabilistic binary classifiers, which enables us to directly inherit the desirable properties of unbiased PU learning.
>
> We believe that, through this contribution, our framework that integrates deep detectors with unbiased PU learning is both novel and practically important for the semi-supervised anomaly detection problem.
>
>
> > The use of an absolute value in the final objective function (Eq. 13) to prevent divergence, while citing prior work, feels somewhat like an ad-hoc fix. This practical necessity might suggest an underlying instability when combining the unbiased PU risk estimator with highly complex models like deep neural networks. The paper would be more impactful if it included a deeper analysis of why this instability occurs and the theoretical implications of this absolute value "patch."
>
> In conventional PU learning with deep models, Kiryo et al. (2017) pointed out that the approximation in Eq. (12) can become negative, which may lead to over-fitting. To address this issue, they introduced a max-based correction: $\max(0,L)$, while Hammoudeh and Lowd (2020) presented an alternative absolute-value correction as in Eq. (13). Both works provide theoretical analyses and show that their estimators are statistically consistent.
>
> In particular, since the absolute-value correction is proven to hold for arbitrary functions, it also yields a consistent estimator in our setting. Furthermore, it has the practical advantage of being simpler to implement than the max-based approach. We will include this discussion in the revised paper.

---

### Official Review · Reviewer_Q8eX · 2025-11-02

**Soundness:** 3
**Presentation:** 3
**Contribution:** 2
**Rating:** 6
**Confidence:** 4

**Summary:**

The authors propose to apply the assumptions of PU learning to outlier detection. They derive a reweighted loss based on the assumption that one knows approximately the contamination rate of the unlabeled data.  They derive a stabilized version of the loss in eq (13).   They apply it to autoencoder loss and to SVDD. They compare it on several datasets against several baselines. They measure experimentally the sensitivity to mismatch of the assumed contamination rate. They evaluate against seen and unseen anomalies.

**Strengths:**

- Clear idea
- Easily readable paper
- Evaluation of different contamination rates
- Evaluation against seen and unseen anomalies
- Application of the idea for more than one baseline
- reasonable performance, limitations are discussed

**Weaknesses:**

- some of the datasets (mnist, fashionmnist, cifar-10) are abit outdated and simplistic as of 2025
- the idea was tried with two rather old baselines. it would be good to try it out with a newer approach

**Questions:**

na

---

> ### Author Response · Authors · 2025-11-21
>
> Thank you for your insightful comments, which we shall address below.
>
>
> > some of the datasets (mnist, fashionmnist, cifar-10) are a bit outdated and simplistic as of 2025
>
> As you commented, MNIST, FashionMNIST, SVHN, and CIFAR-10/100 are relatively old datasets. We used these datasets since they were used in prior work such as (Ruff et al., 2019) and (Qiu et al., 2022). On the other hand, Path, OCT, and Tissue used in Section 5 are real-world medical data, and MVTecAD used in Appendix E is a standard anomaly detection benchmark as of 2025. On these recent and complex datasets, our approach achieves better performance than existing approaches.
>
>
> > the idea was tried with two rather old baselines. it would be good to try it out with a newer approach
>
> Thank you for the helpful suggestion. Our framework can be applied as long as the loss function of the base anomaly detector is non-negative and differentiable. In addition to AE and DeepSVDD used in this paper, it should also be applicable to more recent methods such as [1,2]. We leave the application of our framework to these methods for future work.
>
> [1] Qiu, Chen, et al. "Neural transformation learning for deep anomaly detection beyond images." International conference on machine learning. PMLR, 2021.
>
> [2] Shenkar, Tom, and Lior Wolf. "Anomaly detection for tabular data with internal contrastive learning." International conference on learning representations. 2022.

---

### Author Response · Authors · 2025-11-28

Dear reviewers,

Thank you very much for your constructive and thoughtful reviews. I would greatly appreciate it if you could share any additional comments based on my responses.

Best regards,

---

### Author Response · Authors · 2025-12-02

Dear Area Chairs,

We would like to express our sincere gratitude and appreciation for your work as the area chair. To assist your assessment, we briefly summarize below the main points of the discussion between us and the reviewers.

---

**Reviewer 4853:**
Following our initial response to Reviewer 4853, the reviewer indicated that most of their concerns had been resolved and that they would raise their score from 2 to 4. In their subsequent comment, they further noted that, “with the expectation of seeing the following additional results, I would like to further raise my score.” In response, we conducted the requested additional experiments. These additional results demonstrate that our approach achieves higher performance across a variety of datasets and even when compared with a recent state-of-the-art method presented at ICML 2025. In light of these findings, we believe that we have been able to fully address all of the reviewer’s concerns.

**Reviewer qX5f:**
Following our response to Reviewer qX5f, the reviewer indicated that most of their concerns had been resolved. They also mentioned that they would keep their score unchanged for the moment, but that this might depend on the feedback exchanged with the other reviewers. Since Reviewer 4853 decided to raise their score, we believe this suggests that the concerns raised by the other reviewers have also been adequately addressed.

**Reviewers KjdH and Q8eX:**
Although we did not receive replies from these reviewers, we believe that our responses successfully addressed their concerns regarding the novelty of the method, its theoretical guarantees, and the validity of the experiments. In addition, our response to Reviewer 4853 includes additional comparison results against a recent state-of-the-art method presented at ICML 2025. We believe these results will further convince these reviewers of the effectiveness of the proposed method.

---

After this rebuttal period, we believe that we have been able to address almost all of the reviewers’ concerns. We hope that this summary will be helpful in your final decision-making.

Best regards,

---

### Meta-Review · Area_Chair_CXTg · 2025-12-22

**Summary:**

The remaining major concerns of the three negative reviewers are:  1) the technical novelty and depth of the work are not significant; 2) the experimental setup hasn't been fully explained and justified (e.g., the setting of $\alpha$ and the presence of a validation set).

There is a major limitation that hasn't been discovered by the reviewers. The paper studied the problem of anomaly detection when the unlabeled set is contaminated by anomalies. This means we can use outlier detection or anomaly detection methods to detect the outliers in the unlabeled training set and then conduct classical semi-supervised or even supervised anomaly detection. Such naive baselines haven't been compared in the experiments; most of the baselines compared in the experiments are sensitive to outliers. So there is no evidence that the studied problem is really non-trivial or challenging. I suggest that the authors add the corresponding experiments in the next version of the paper, which is also useful to support the novelty and rationality of combining PU learning with Deep AD.

**Reviewer Concerns:**

**Reviewer Q8eX (6):**
1. Some datasets used in the experiments are a bit outdated or simplistic.
2. The idea was tried with two rather old baselines.

I think these are minor issues and do not hamper the contribution of the work.

**Reviewer KjdH (4):**
1. The technical novelty is limited: application of a well-established theoretical framework.
2. The theoretical depth should be strengthened: e.g., the absolute value in (13) hasn't been clearly analyzed.

I think the second point has been partially addressed by the rebuttal, while the first one remains.

**Reviewer qX5f (4):**
1. Limited Technical Novelty: a direct application of the unbiased PU learning framework from Kiryo et al. (2017).
2. Missing α Estimation Mechanism.
3. Incomplete Experimental Validation.
4. Limited Performance Improvement.

I think the rebuttal has addressed weaknesses 2 and 3, at least partially.''

**Reviewer 4853 (2):**
1. The novelty is limited.
2. No theoretical analysis is provided.
3. The baselines are not novel enough.
4. The evaluation is merely on image datasets.
5. The rationality of the validation set is questionable.
6. The hyperparameter analysis is insufficient.
7. The experimental setup is weak.

I think concerns 3, 6, and 7 have been addressed by the rebuttal and revision, while other concerns remain.

**Reviewer Scores:**

During the rebuttal, Reviewer 4853 was willing to raise the score from 2 to 4, and Reviewer qX5f stated that he/she kept the score 4 unchanged temporarily. It is unlikely that the three negative reviewers would increase their ratings to 6 or higher even after a full discussion, as they share a common concern regarding the novelty and technical depth of the work.

---

### Decision · Program_Chairs · 2026-01-26

Reject